# Asymmetric elastoplasticity of stacked graphene assembly actualizes programmable untethered soft robotics

Shuai Wang [1,2,6], Yang Gao [3,4,6], Anran Wei [3,6], Peng Xiao [1,2✉], Yun Liang[1,2], Wei Lu [1,2], Chinyin Chen[5], Chi Zhang [5], Guilin Yang[5], Haimin Yao [3,4✉] & Tao Chen [1,2✉]

There is ever-increasing interest yet grand challenge in developing programmable untethered soft robotics. Here we address this challenge by applying the asymmetric elastoplasticity of stacked graphene assembly (SGA) under tension and compression. We transfer the SGA onto a polyethylene (PE) film, the resulting SGA/PE bilayer exhibits swift morphing behavior in response to the variation of the surrounding temperature. With the applications of patterned SGA and/or localized tempering pretreatment, the initial configurations of such thermal-induced morphing systems can also be programmed as needed, resulting in diverse actuation systems with sophisticated three-dimensional structures. More importantly, unlike the normal bilayer actuators, our SGA/PE bilayer, after a constrained tempering process, will spontaneously curl into a roll, which can achieve rolling locomotion under infrared lighting, yielding an untethered light-driven motor. The asymmetric elastoplasticity of SGA endows the SGA-based bi-materials with great application promise in developing untethered soft robotics with high configurational programmability.

[1] Key Laboratory of Marine Materials and Related Technologies, Zhejiang Key Laboratory of Marine Materials and Protective Technologies, Ningbo Institute of Materials Technology and Engineering, Chinese Academy of Sciences, 315201 Ningbo, People's Republic of China. [2] School of Chemical Sciences, University of Chinese Academy of Sciences, 19A Yuquan Road, 100049 Beijing, People's Republic of China. [3] Department of Mechanical Engineering, The Hong Kong Polytechnic University, Hung Hom, Kowloon, Hong Kong SAR, People's Republic of China. [4] The Hong Kong Polytechnic University Shenzhen Research Institute, 518057 Shenzhen, People's Republic of China. [5] Zhejiang Key Laboratory of Robotics and Intelligent Manufacturing Equipment Technology, Ningbo Institute of Materials Technology and Engineering, Chinese Academy of Sciences, 315201 Ningbo, People's Republic of China. [6] These authors contributed equally: Shuai Wang, Yang Gao, Anran Wei. ✉email: xiaopeng@nimte.ac.cn; mmhyao@polyu.edu.hk; tao.chen@nimte.ac.cn

nspired by nature, scientists have begun to explore the approaches to manufacturing and controlling soft robotics, which is made from flexible materials and therefore can safely interact with living organisms or fragile objects, bridging the gap between robots and human beings[1–3]. With their structural deformability and diverse responsive materials, soft robotics can achieve complex morphing behavior in response to a variety of external stimuli, forming desired geometries, bearing mechanical loadings, and performing propulsion, and actuation. Current soft robotics mainly relies on pneumatic networks embedded in elastomeric rubbers, and therefore most of them should be tethered to the external power sources and control systems. This greatly constrains their applications in practices[4]. Developing soft smart materials that could convert external energy such as thermal, light, or chemical to the mechanical energy so as to achieve controllable morphing actuation should be of great value to the development of untethered soft robotics.

Recently, a variety of flexible morphing systems have been developed with configurable soft materials by taking the advantage of various mechanisms such as asymmetric thermal expansion[5,6], liquid crystalline transitions[7,8], phase transitions[9–12], and anisotropic swelling[13–21], etc. To achieve shape programming and customization, numerous efforts have been made by applying specific chemical structures through a variety of polymeric materials including shape memory polymers (SMPs)[22–25], vitrimer[26], hydrogel[27–29], organogel[30]. Application of two-way SMPs with a heterogeneous semi-crystalline structure or a broad melting transition realized reversible and controllable morphing behaviors[31–33]. Exchangeable bonds and photocrosslinking were utilized in liquid crystal polymer networks to spatially organize the actuating mono-domains, resulting in programmable shape design[34–38]. Temperature-responsive hydrogels with spatially and temporally controllable expansion and contraction were fabricated to achieve desired 3D structures and programmed motions[39–41].

In order to apply these smart materials in wearable and/or field robotics, the essential components including processing, actuation, and power should be fully integrated and embedded in its own structure. This poses a grand challenge in creating untethered soft robotics[42]. Up to now, several frontier attempts have been made successfully in this regard. For example, Daraio and Lewis et al.[43] reported soft robotics composed of liquid crystal elastomer (LCE) bilayers with orthogonal director alignment and different nematic-to-isotropic transition temperatures to form active hinges that interconnect polymeric tiles. The printed LCE hinges exhibit a reversible bending deformation upon heating to a temperature above their actuation temperatures. By using a hygroscopically responsive film consisting of aligned nanofibers, Kim et al.[44] developed an untethered soft humidity-powered robot that can locomote spontaneously in a ratcheted fashion on a moist surface. Despite these successes, the morphing systems used in untethered soft robotics still have limitations such as the complicated procedure of synthesis, lower programmability, and slow responding speed due to the time-dependent working mechanisms such as diffusion and molecular organization. Creating novel smart materials with the facile manufacturing process, high programmability and high actuation performance should be of great value to the development of untethered soft robotics.

As one of the well-known two-dimensional materials, graphene exhibits ultrahigh flexibility which makes it an excellent candidate for the next-generation soft actuator[45–47]. Our study on stacked graphene assembly (SGA) revealed that it exhibits high plasticity under tension and high elasticity under compression. Based on this finding of asymmetric elastoplasticity, here we develop a strategy to prepare a cost-effective and highly programmable

smart material by transferring a layer of SGA onto a polyethylene (PE) film. The resulting SGA/PE bilayer film exhibits sensitive morphing behaviors in response to the variation of the surrounding temperature. Specifically, it will curl with the SGA layer wrapped inside as the environmental temperature increases and flatten when the temperature is reduced to the initial value (Fig. 1a, d). More interestingly, if an as-prepared SGA/PE bilayer is pre-treated with a heating and a subsequent cooling process (similar to the tempering process in metallurgy) in a constrained space, it will coil spontaneously into a roll with PE layer wrapped inside once the constraint is removed (Fig. 1a–c). The curvature of the resulting roll can be programmed by controlling the tempering temperature. Compared with the untreated SGA/PE film, a surprising finding is that the resulting roll exhibits an opposite morphing behavior in response to temperature variation (Fig. 1e). By changing the thickness, layout, and distribution of the SGA film or applying nonuniform and localized tempering, more sophisticated 2D and 3D structures with reversible morphing capability can be realized with the SGA/PE bilayer. Such high programmability of the SGA/PE bilayer is essentially attributed to the asymmetric elastoplastic properties of SGA film (Fig. 1f) as demonstrated by molecular dynamics (MD) simulation and confirmed by finite element simulations. Thanks to the programmable morphing behavior, the SGA/PE bilayer is applied to construct a variety of actuation systems such as sit-up robots, artificial iris, artificial water lily (Fig. 1g). More importantly, different from normal bilayer actuators, the SGA/PE bi-material can spontaneously curl into a roll, which even achieves rolling locomotion under lateral infrared (IR) lighting, yielding an untethered light-driven motor as soft robotics (Fig. 1h). Asymmetric elastoplasticity of SGA endows the SGA/PE bilayer materials with great application potential in actuators, motors, and untethered soft robotics with high programmability.

## Results

**Fabrication and characterization of SGA/PE bilayer films**. Figure 2a schematically shows the fabrication process of SGA/PE bilayer using the Langmuir–Blodgett (LB) method[48–50], which starts from spraying a certain amount of graphene/ethanol dispersion onto the surface of pure water. A piece of sponge is gradually inserted into the water from aside. Due to the change of surface tension, the floating graphene flakes recede away from the sponge and aggregate to form a condensed SGA layer. A piece of PE strip backed by a glass slide is used to transfer the SGA from the water surface to the PE layer. An SGA/PE bilayer film is obtained after drying in nitrogen gas at room temperature (Fig. 2b). Multi-ply SGA film can be prepared by repeating the transferring process for multiple times. The obtained SGA/PE bilayer film can be tailored to specific dimensions and shapes as required. Figure 2c shows the cross-sectional scanning electronic microscopy (SEM) image of SGA/PE bilayer. A top view SEM image of the SGA film shows that the staggered graphene sheets form a closely stacked structure (Fig. 2d). The thickness of the SGA layer (six plies) is about 530 nm (Fig. 2e). The Raman spectrum of the SGA film and PE film is shown in Fig. 2f. The characteristic D and G bands of graphene occur at about 1346 and 1576 cm$^{-1}$ respectively. The small intensity ratio between the D band and G band ($I_D/I_G = 0.7$) confirms the good quality of the graphene layer. The Raman spectrum of the PE layer shows sharp characteristic peaks at 1063, 1130, 1305, 1418, 1460 cm$^{-1}$, implying the presence of crystalline regions in the PE layer[51]. Figure 2g shows the stress–strain curves of PE film measured along the alignment direction (AD) and transverse direction (TD), respectively. Clearly, PE film exhibits higher strength ($S = 22$ MPa) but lower ductility (%EL = 200%) along the AD

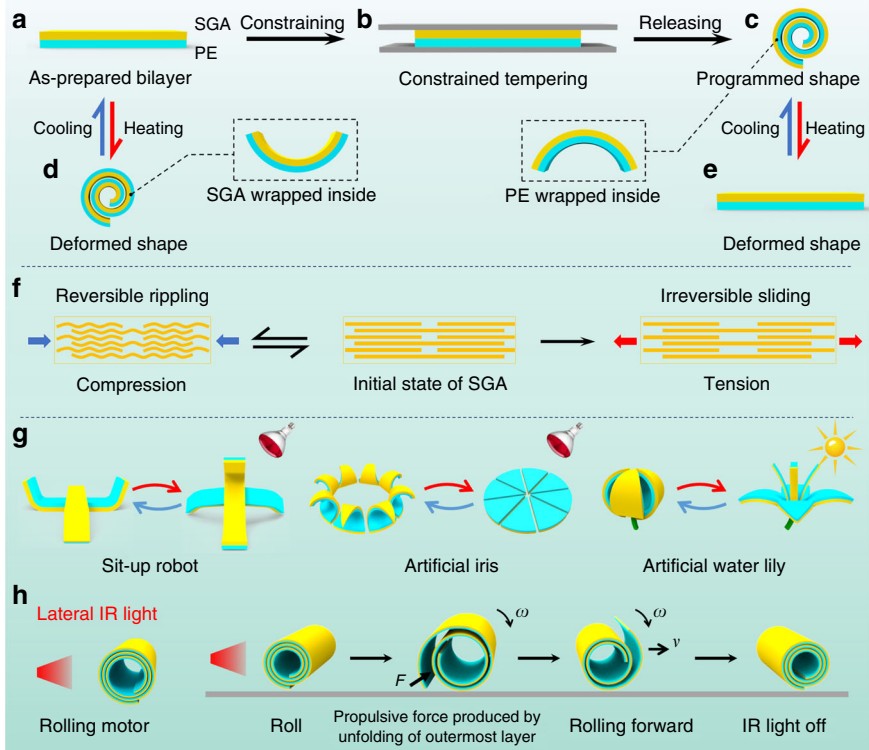

**Fig. 1 Illustration of programmable thermal-induced morphing systems based on SGA/PE bilayer films. a–c** Schematics showing the process of constrained tempering on SGA/PE films. **d, e** Schematics showing the thermal-induced shape morphing of an as-prepared SGA/PE film and a tempered SGA/PE film respectively. **f** Schematic illustration of the deformation mechanism accounting for the asymmetric elastoplasticity of the SGA under tension and compression. **g** Three representative shape-morphing systems made of SGA/PE bilayers with programmable configurations. **h** SGA/PE bilayer roll as a light-driven rolling motor.

compared to those along the TD ($S = 13$ MPa, %EL = 1600%). Such high mechanical anisotropy of the PE film is essentially attributed to the preferred orientation of the PE molecules, which is caused by the directional stretching in the blow molding process of manufacturing. Such strain-induced crystallization in the PE film is also confirmed by the symmetric 2D-XRD pattern (Fig. 2h). Due to the existence of the oriented crystallinity in PE, the SGA/PE bilayer film will always bend about the transverse axis under thermal stimuli no matter the long side of the sampling box is parallel or perpendicular to the AD of the PE film (Fig. 2i).

**Thermal-induced morphing behaviors of SGA/PE bilayer film.** The as-prepared flat SGA/PE bilayer, when heated, will bend and curl into a roll. This is basically attributed to the misfit of thermal strains between the SGA and PE layers. The thermal expansion coefficient of graphene[52] is almost neglectable in comparison to that of PE ($4.0 \times 10^{-4}$/°C)[53]. When temperature increases, the resulting eigenstress on the SGA/PE interface produces bending moments, making the bilayer film to curl. Under this circumstance, the SGA layer is wrapped inside the PE layer, as shown in Fig. 3a. Such thermal-induced morphing behavior is recoverable when the temperature returns to the original value and even can be reversed upon further cooling (Supplementary Fig. 1). The actuation curvature increases with the actuating temperature and the SGA ply number while varies little with the size of the films (Supplementary Figs. 2, 3).

A more interesting morphing behavior can be observed from an SGA/PE film that has been pre-tempered in a constrained space, as shown in Fig. 3b. Here, an as-prepared flat SGA/PE bilayer film is sandwiched by two glass slides. Heat treatment is

then performed by increasing the temperature by $\Delta T$ and then cooling down to the room temperature (like the tempering process in metallurgy). After releasing the constraint of the glass slides, the SGA/PE film curls spontaneously into a roll with curvature dependent on the ply number of the SGA layer and the tempering temperature ($\Delta T$), as shown in Fig. 3c, d. Surprisingly, now the PE layer is wrapped inside by the SGA layer, which is opposite to that of the as-prepared sample without tempering (Fig. 3a). Such opposite curling direction can be attributed to the asymmetric elastoplastic property of the SGA layer under tension and compression (see the next section for elaboration). The size and aspect ratio are demonstrated to have little effect on the curvatures of the tempered SGA/PE films (Supplementary Fig. 4). Nevertheless, the curvature of a tempered device varies with the applied actuating temperature (Supplementary Fig. 5). The tempering effect would be greatly weakened if the film is constrained in the in-plane direction as well, which also confirms the importance of the asymmetric elastoplasticity of the SGA to the shape programmability of the SGA/PE bilayer (Supplementary Fig. 6). The SGA/PE roll produced by tempering, when heated, will flatten in 0.37 s (Fig. 3e) and recover to the curled configuration in 1 s upon cooling. This process can be repeated with high reversibility, as indicated by the stable variation of the equivalent curvature shown in Fig. 3f. Even after 1000 cycles, no noticeable deterioration is observed in the actuation performance of the film (Supplementary Fig. 8 and Supplementary Movie 1), indicating the high stability of such unrolling-rolling actuation. Such morphing behavior is conserved even after the actuator is trampled along different directions (Supplementary Fig. 9). Compared to the other actuators reported in literature[6,14,16,17,19,44,54–59], our SGA/PE roll stands out in the morphing rate ($0.089\,\text{s}^{-1}$), as shown in Fig. 3g and

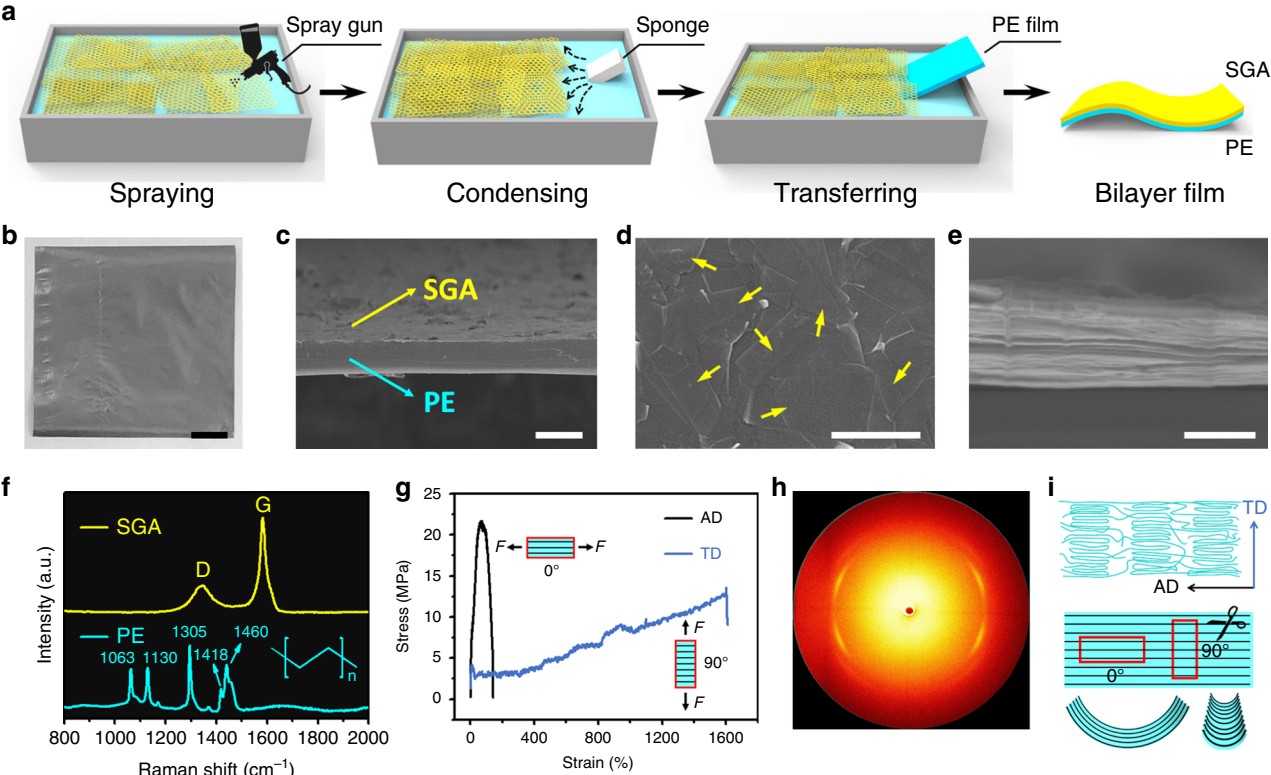

**Fig. 2 Fabrication and characterizations of SGA/PE bilayer films. a** Schematic illustration of the fabrication process of SGA/PE bilayer films using the L-B method. Graphene flakes are sprayed on the water surface and condensed by the change of surface tension induced by the insertion of a piece of sponge. The SGA film floating on the water surface is then transferred onto a PE film and dried at room temperature, giving rise to a flexible SGA/PE bilayer film. The cost of materials applied in the whole process is ~\$0.0024 cm$^{-2}$. **b** Optical image of an SGA/PE bilayer film with the SGA layer on the top side. The film exhibits gray color on the SGA side while black color on the PE side (not shown here). **c** Cross-sectional SEM image of an SGA/PE bilayer film. The top layer is SGA and the bottom layer is PE. **d** SEM image of the surface of an SGA film. **e** Cross-sectional SEM image of an SGA film. **f** Raman spectra of an SGA film and a PE film. **g** Stress–strain curves of PE along the alignment direction and transverse direction, respectively. AD alignment direction, TD transverse direction. **h** 2D-XRD pattern of a PE film shows bright spots, implying the presence of crystalline structures in the PE film. **i** Schematic illustration of the strain-induced alignment of the PE molecules and the bent configurations when composited with SGA film in response to the external thermal stimuli. The sampling boxes are illustrated by red rectangles. Scale bars: 1 cm (**b**), 20 μm (**c**), 2 μm (**d**), 500 nm (**e**).

Supplementary Table 1. Moreover, as an actuator, the SGA/PE roll can lift a load of ten times of its own weight upon controlled IR light illumination with an intensity of 120 mW cm$^{-2}$ (Supplementary Fig. 10).

**Asymmetric elastoplasticity of SGA.** Figure 3b shows that a constrained tempering process could make an SGA/PE bilayer curl spontaneously into a roll after releasing, while the curling direction is opposite to that of a pristine SGA/PE bilayer film under heating (Fig. 3a). This implies that the residual stress after the constrained tempering is compressive in the SGA layer and tensile in the PE layer. Therefore, residual elongation must exist in the SGA layer after the tempering process, during which it is firstly stretched and then compressed by the bonded PE layer in response to the temperature variation. That is, the elongation of the SGA layer during the heating stage has not been fully recovered by the compression experienced in the cooling stage. Such mechanical behavior of the SGA is essentially attributed to its asymmetric elastoplastic behavior under tension and compression. As testified by MD simulations, SGA exhibits high plasticity under tension while high elasticity under compression (Fig. 4a). The plasticity under tension is essentially due to the irreversible sliding between the graphene flakes, while the elasticity under compression results from the reversible rippling-like deformation[60] at the nanoscale (Fig. 4b and Supplementary Movie 2). For effective tempering, plastic deformation in the SGA

layer is necessary which requires a minimum temperature increment (see Supplementary Note 1)

$$\Delta T^{*} = \frac{S_{\mathrm{SGA}}^{\mathrm{t}}}{\alpha(1 + \nu_{\mathrm{PE}})\sqrt{1 - \nu_{\mathrm{SGA}} + \nu_{\mathrm{SGA}}^{2}}} \cdot \left(\frac{1}{E_{\mathrm{SGA}}^{\prime\mathrm{t}}} + \frac{t_{\mathrm{SGA}}}{t_{\mathrm{PE}}E_{\mathrm{PE}}^{\prime}}\right), \quad (1)$$

where $E_{\mathrm{SGA}}^{\prime\mathrm{t}} = E_{\mathrm{SGA}}^{\mathrm{t}}/(1 - \nu_{\mathrm{SGA}}^{2})$ and $E_{\mathrm{PE}}^{\prime} = E_{\mathrm{PE}}/(1 - \nu_{\mathrm{PE}}^{2})$, with $E_{\mathrm{SGA}}^{\mathrm{t}}$, $S_{\mathrm{SGA}}^{\mathrm{t}}$, $\nu_{\mathrm{SGA}}$, $t_{\mathrm{SGA}}$ being the tensile elastic modulus, tensile yield strength, Poisson's ratio, and thickness of the SGA layer respectively, and $E_{\mathrm{PE}}$, $\nu_{\mathrm{PE}}$, $\alpha$, and $t_{\mathrm{PE}}$ stand for the elastic modulus, Poisson's ratio, thermal expansion coefficient and thickness of the PE layer, respectively. Taking $E_{\mathrm{SGA}}^{\mathrm{t}} = 20.7\,\mathrm{GPa}$, $S_{\mathrm{SGA}}^{\mathrm{t}} = 20.4\,\mathrm{MPa}$, $\nu_{\mathrm{SGA}} = 0.19$, $t_{\mathrm{SGA}} = 0.3\,\mu\mathrm{m}$, $E_{\mathrm{PE}} = 300\,\mathrm{MPa}$[61], $\nu_{\mathrm{PE}} = 0.46$[62], $\alpha = 4 \times 10^{-4}/{}^{\circ}\mathrm{C}$[53], and $t_{\mathrm{PE}} = 10\,\mu\mathrm{m}$, estimation based on Eq. (1) indicates that $\Delta T^{*} \approx 4.8\,{}^{\circ}\mathrm{C}$. When $\Delta T > \Delta T^{*}$, the curling curvature of the tempered SGA/PE bilayer, $\kappa$, is given by[63] (see Supplementary Note 1):

$$\kappa \approx \frac{6\alpha(1 + \nu_{\mathrm{PE}})(\Delta T - \Delta T^{*})}{4t_{\mathrm{PE}} + E_{\mathrm{PE}}^{\prime}t_{\mathrm{PE}}^{2}/E_{\mathrm{SGA}}^{\prime\mathrm{c}}t_{\mathrm{SGA}}}, \quad (2)$$

where $E_{\mathrm{SGA}}^{\prime\mathrm{c}} = E_{\mathrm{SGA}}^{\mathrm{c}}/(1 - \nu_{\mathrm{SGA}}^{2})$, and $E_{\mathrm{SGA}}^{\mathrm{c}}$ denotes the compressive elastic modulus of the SGA layer. The dependence of $\kappa$ on $t_{\mathrm{SGA}}$ given by Eq. (2) is plotted in Fig. 4c, which is in good consistency with the experimental measurements as shown in Fig. 3d. Such spontaneous curling behaviors of the SGA/PE bilayer after constrained tempering can be reproduced by finite

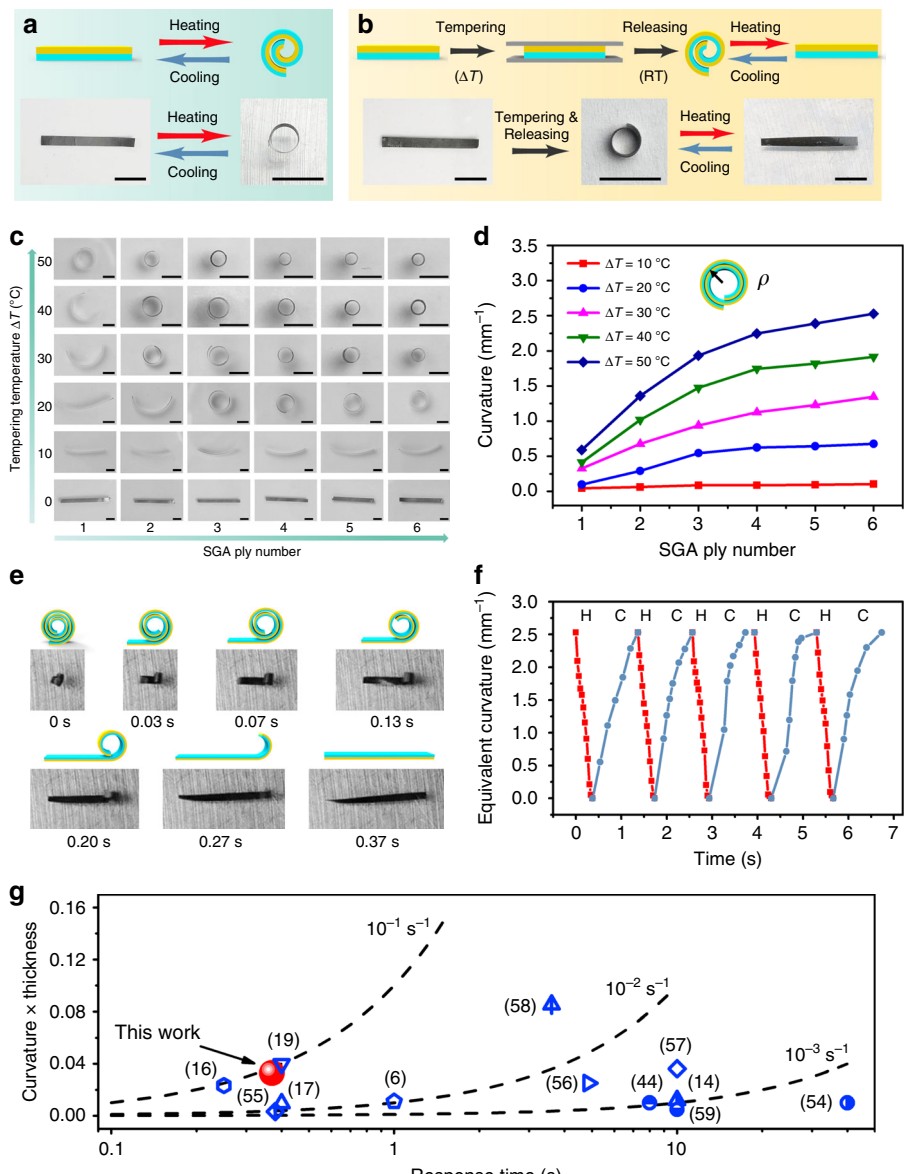

**Fig. 3 Thermal-induced morphing behaviors of SGA/PE films. a** Schematic illustration and optical images showing reversible morphing behavior of an as-prepared flat SGA/PE film upon heating ($\Delta T = 10\,^{\circ}\text{C}$) and cooling to room temperature. **b** Schematic illustration and optical images of constrained tempering on an SGA/PE bilayer and the reversible morphing behavior exhibited by the tempered SGA/PE bilayer in response to thermal stimuli. **c** Optical images showing the deformed configurations of SGA/PE bilayers with different SGA ply numbers (1, 2, 3, 4, 5, and 6) after being tempered by different temperature changes ($\Delta T = 0$, 10, 20, 30, 40, and 50 °C). **d** Dependence of curling curvature of tempered SGA/PE bilayer on the SGA ply number for different tempering temperatures. Here, the curling curvature is defined as reciprocal of the radius ($\rho$), which is the distance from the arc center to the middle of the wall. **e** Snapshots showing the uncurling process of an SGA/PE roll in response to temperature change ($\Delta T = 40\,^{\circ}\text{C}$). **f** Variation of the equivalent curvature of an SGA/PE roll with time upon a cyclic exposure to a thermal stimulus ($\Delta T = 40\,^{\circ}\text{C}$). Here equivalent curvature is defined as the curvature of an equivalent complete roll (see Supplementary Fig. 7 for details). H heating, C cooling. **g** Comparison of morphing performance between our SGA/PE roll (red sphere) and other morphing systems reported in the literature (blue symbols, see Supplementary Table 1 for data) on a "curvature × thickness" vs. response time plane. The dash lines here represent the contours of constant actuation rates with "curvature × thickness"/response time = 0.1, 0.01, 0.001 s$^{-1}$, respectively. Scale bars: 5 mm (**a**, **b**); 2 mm (**c**).

element-based simulations (Supplementary Fig. 11 and Supplementary Movie 3), in which the SGA is modeled as a continuum with asymmetric elastoplastic behavior as revealed by MD simulation (Fig. 4a).

**SGA/PE-based morphing systems with programmable configurations.** The asymmetric elastoplastic behavior of SGA enables us to program the initial configurations of the SGA/PE-based morphing systems through different strategies. Figure 5a

illustrates some examples of the programmed configurations achieved by the strategy of discrete SGA patches under uniform constrained tempering. For instance, three discrete SGA patches on one side of a PE film result in a "V" shape (Fig. 5a, left). Alternately distributed SGA patches on both sides of a PE film give rise to a wavy shape (Fig. 5a, right). If the discrete SGA patches are allocated along an oblique direction relative to the side of the PE layer, spiral, and helical geometries will be produced (Fig. 5b). Applying an SGA patch with gradient thickness on one side of the PE film produces a spiral structure with varying

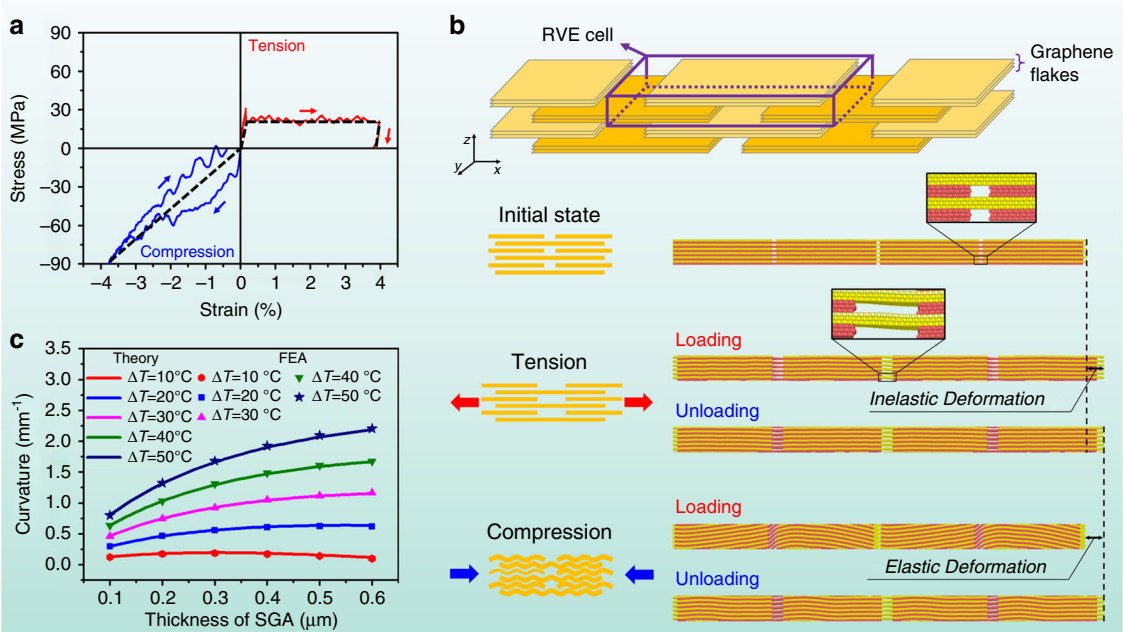

**Fig. 4 Numerical verification of the asymmetric elastoplasticity of SGA. a** The loading and unloading stress–strain curves of an SGA layer under uniaxial tension (red curve) and compression (blue curve) as calculated by MD simulation. The black dash lines represent the equivalent stress–strain relations of SGA when treated as a continuum. **b** Schematics of the idealized structure of SGA and the calculated snapshots of the deforming process under tension and compression by MD simulation (see Supplementary Movie 2 for details). **c** Variation of curling curvature ($\kappa$) of tempered SGA/PE bilayer as a function of SGA layer thickness and tempering temperature as predicted by theory (Eq. 2) and FEA simulation. Here, the following typical values are adopted: $E^t_{SGA} = 20.7$ GPa, $E^c_{SGA} = 2.2$ GPa, $S^t_{SGA} = 20.4$ MPa, $\nu_{SGA} = 0.19$, $t_{SGA} = 0.1 \sim 0.6$ $\mu$m, $E_{PE} = 300$ MPa[61], $\nu_{PE} = 0.46$[62], $\alpha = 4 \times 10^{-4}/°C$[53], and $t_{PE} = 10$ $\mu$m.

curvature (Fig. 5c). Applying PE film with the long edge at a certain angle to its alignment direction produces a 3D helical structure (Fig. 5d). All these geometries can be successfully predicted by the finite element-based simulations. A combination of the above strategies can produce 3D conical helical structures (Supplementary Fig. 12a) and complex patterns and shapes (Supplementary Fig. 12b).

Another strategy for programming the initial configuration is to apply nonuniform or localized heating in the constrained tempering process (Fig. 5e). If the constrained tempering is carried out only on one side or both two sides of the SGA/PE, curling takes place on these pre-set sides only. Applying the gradient temperature field during tempering leads to nonuniform curling curvature along the SGA/PE film. A more sophisticated configuration can be obtained through a combination of the above-mentioned strategies (Supplementary Fig. 12c–e). In the above cases, the nonuniform heating is achieved with sliced glass slides on a hotplate (see "Methods" section). Alternatively, more localized tempering can be achieved through direct laser writing to achieve more sharp bending. For example, laser writing along a line on the constrained SGA/PE bilayer will create a fold with controllable folding angle dependent on the ply number of SGA and irradiation time of laser (Fig. 5f), size of the laser beam (Supplementary Fig. 13a–c), and writing path (Supplementary Fig. 13d). The sequential application of such laser writing can turn a planar SGA/PE sheet into a 3D box (Fig. 5g), which shows the competence of constrained tempering of SGA/PE in shape programming. The cube keeps its configuration at ambient temperature and unfolds upon heat ($\Delta T = 10$ °C) (Supplementary Fig. 14).

**SGA/PE-based morphing actuators**. Thanks to the programmable initial configurations of the SGA/PE bilayer, the SGA/PE films can be applied to generate actuators with complex shapes.

Figure 6 shows two types of basic units made from SGA/PE bilayer film including curved unit and folded unit. These units can be assembled as needed to produce more complicated actuation systems. An artificial water lily (Fig. 6a, b and Supplementary Movie 4) is assembled from the curved units. The flower is initially in bud and blooms in less than 2 s upon the exposure to natural sunlight (20 mW cm$^{-2}$), resembling the natural water lily which blooms in the daytime and closes at night. Besides the sunlight, the artificial flower also responds to IR light and blooms even faster (<1.5 s) upon IR light exposure (Supplementary Fig. 15). Moreover, an artificial iris is produced by assembling the curved units. It can spontaneously adjust the aperture according to the light illumination (Supplementary Fig. 16 and Supplementary Movie 5). Two folded units are assembled to form a simple toy robot to achieve a reversible light-driven action from lying-down status to sitting-up status in about 1 s (Fig. 6c, d and Supplementary Movie 6). Additionally, the tempered SGA/PE bilayers exhibit great application potential in information storage, encryption, and decryption as shown in Supplementary Fig. 17.

**SGA/PE-based untethered motors**. More importantly, different from normal bilayer actuators, the SGA/PE bilayer can also achieve untethered locomotion under the controllable illumination of IR light. As shown in Fig. 7a and Supplementary Movie 7, when an SGA/PE bilayer roll, which is obtained by a constrained tempering process as introduced above, is illuminated by a lateral IR light on the left side (observed from the end with clockwise outer-to-inner winding), the rolling motion may be triggered. The driving force of such rolling motion is basically attributed to the localized heating by the IR light illumination, which would make the outermost layer of the roll (called "propeller" hereinafter) unfold, flatten, and interact with the ground. The reaction force from the ground pushes the roll to move forward. When the

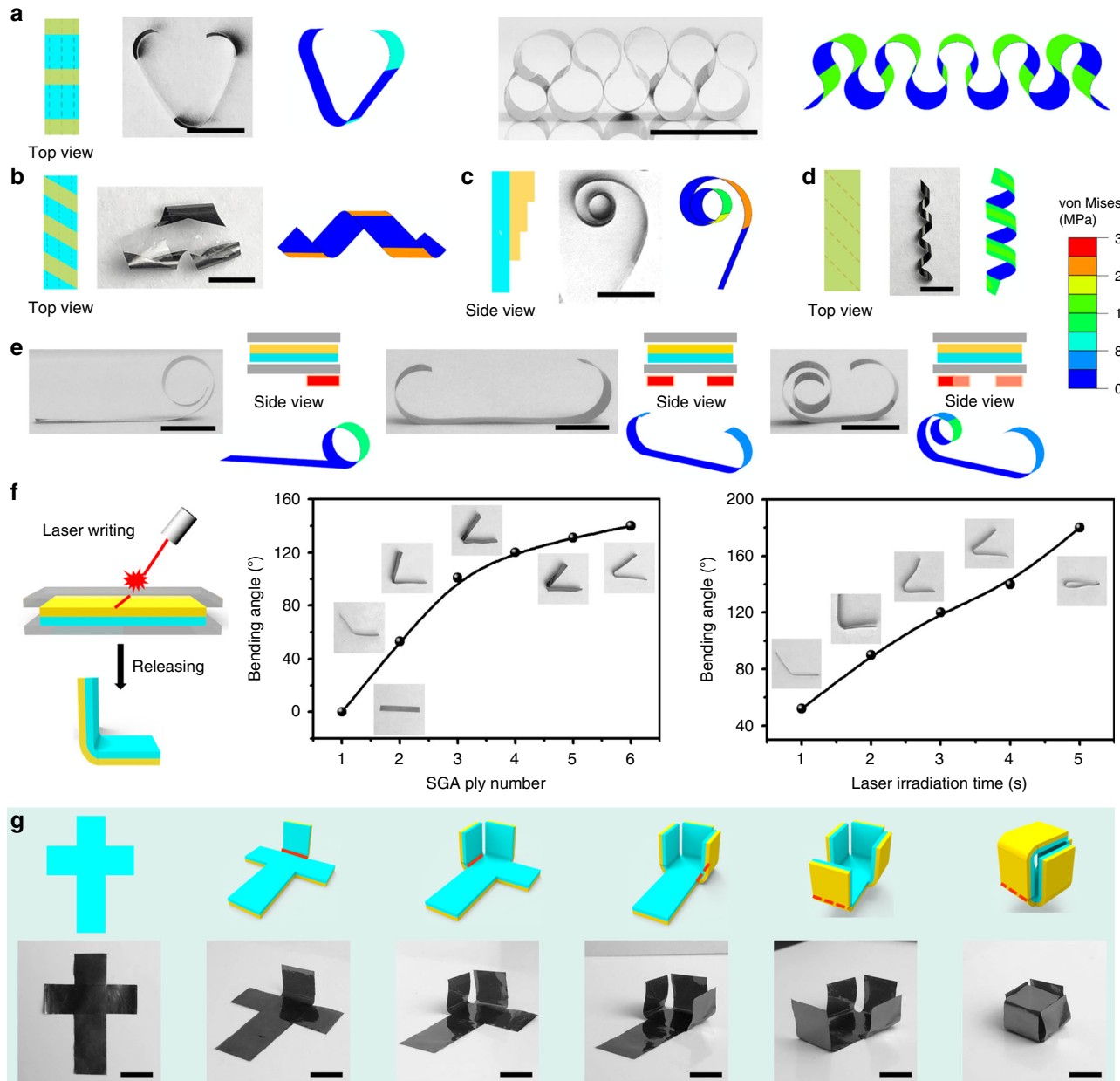

**Fig. 5 Programmable configurations of SGA/PE-based morphing systems achieved by various strategies. a–d** Programmed configurations obtained by customizing SGA or PE. Discrete SGA patches on one side of PE film result in a V-shaped structure (**a** left); SGA patches alternately distributed on both sides of PE film give rise to in a wavy shape (**a** right); Discrete SGA patches allocated along an oblique direction with respect to the long side of the PE layer give rise to a twisting geometry (**b**); SGA patch with gradient thickness produces a spiral (in-plane) (**c**); Continuous SGA layer on a PE layer with alignment direction (denoted by the dash lines) unparallel to the sides produces a 3D helical ribbon (**d**). The right figure in each case is the configuration calculated by FEA simulation. The dash lines in **a**, **b** and **d** represent the alignment direction of the PE layer. **e** Programmed configurations achieved by localized and gradient tempering. The lower right figure in each case is the configuration predicted by FEA simulation. **f** More localized tempering can be achieved through direct laser writing, giving rise to severer bending. The folding angle is dependent on the ply number of SGA and irradiation time of the laser. **g** A planar SGA/PE sheet folds itself into a 3D box after a sequential constrained tempering process with laser writing. Scale bars: 5 mm (**a–e**); 10 mm (**g**).

propeller moves to the back (right) side, its temperature drops due to the loss of IR illumination and the propeller returns to its initial curled status. The momentum of the roll brings the propeller to the front (left) side, where it is subjected to IR illumination again. The driving process mentioned above is repeated, making the roll to move continuously. Here the direction of IR illumination plays a dominant role in making the roll to move. Vertical illumination can cause uncoiling, rather than rolling, of the roll (Fig. 7b). Once the rolling is triggered by lateral IR illumination, the ever-increasing speed is shown in Fig. 7c. There is

an asymptotic upper limit of the rolling speed which, for our system, is estimated to be around 7.19 cm s$^{-1}$ (see Supplementary Note 2). The direction of rolling can be altered by changing the location of illumination along the axis of the roll (Supplementary Fig. 18 and Supplementary Movie 8). It should be pointed out that the occurrence of such rolling motion still relies on the number of winding turns and the diameter of the roll (see Fig. 7d). Rolls with too few turns or too large diameters will uncoil to a flat bilayer under illumination. In addition to rolling on a flat surface, the roll also exhibits the ability to cross obstacles

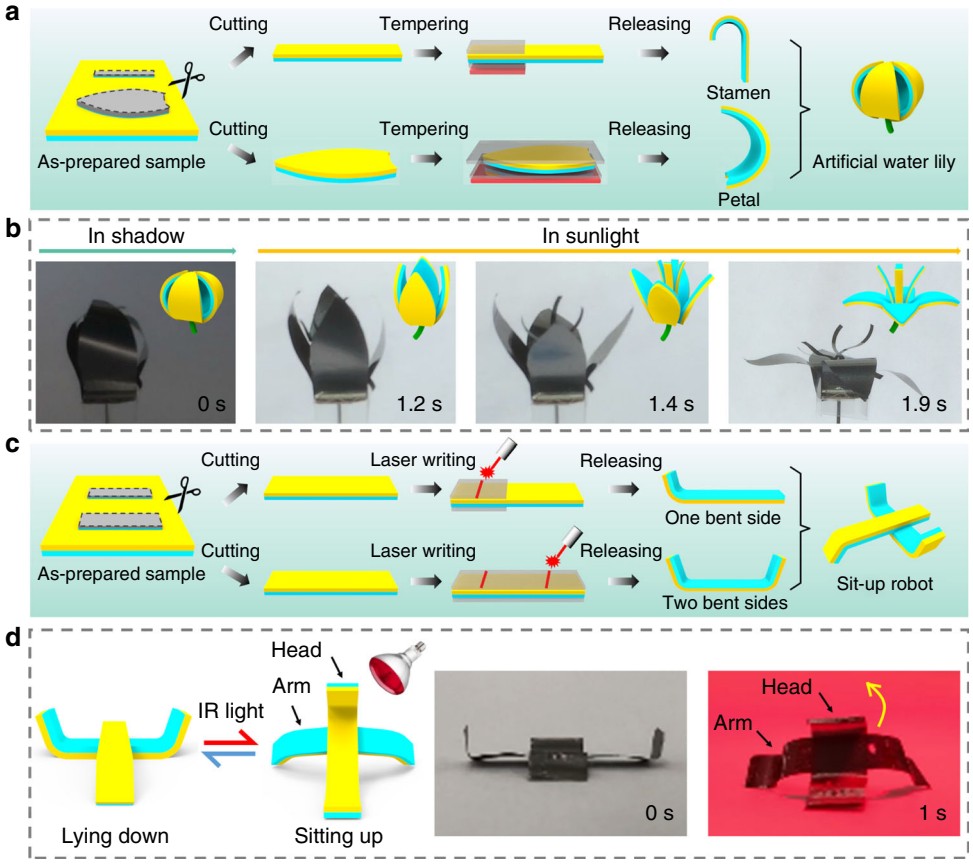

**Fig. 6 Typical morphing systems assembled from basic morphing units of SGA/PE bilayer. a** Schematic illustration of the assembly units of an artificial water lily through constrained tempering. **b** An artificial water lily assembled from the curved units blooms under sunlight (20 mW cm$^{-2}$) (Supplementary Movie 4). **c** Schematic illustration of the assembly units of sit-up robot through constrained tempering. **d** A simple robot assembled from folded units sits up under IR light illumination (Supplementary Movie 6).

with height equal to its radius (Supplementary Fig. 19a), climb ramp with an inclined angle of 10° (Supplementary Fig. 19b) and roll forward on wavy sandy ground (Fig. 7e and Supplementary Movie 8).

The SGA/PE motors can be used to push a toy football into a goal (Fig. 8a and Supplementary Movie 9). The SGA/PE roll and football toy initially stay at the left backcourt. Once the lateral IR light illumination on the central part of the roll is applied, the SGA/PE motor rolls forward and pushes the football toy towards the middle circle of an artificial football field. Then the lateral IR light illumination on the right part of the roll is applied, the SGA/PE motor turns left and pushes the football toy towards the goal of the artificial football field. Finally, the SGA/PE motor pushes the football toy into the goal under continuous lateral IR light illumination. Loading, transporting, unloading of cargoes by an SGA/PE roll under controlled IR light illumination are also demonstrated in Fig. 8b and Supplementary Movie 10. The hollow SGA/PE roll uncoils under vertical IR light illumination. As the light is turned off, the SGA/PE curls and wraps the plastic rod (mimicking the cargo) into the roll. Then, under lateral IR light illumination, the roll with the loaded cargo will move forward to the unloading area. When the SGA/PE motor arrives at the destination, the vertical IR light illumination is applied, and the SGA/PE motor uncoils to unload the cargo. The unfolded SGA/PE roll recovers to the initial curled state once the IR light illumination is off.

Due to the high mobility and loading capacity, the hollow body of the SGA/PE rolls further enables us to assemble two rolls with a shaft (Fig. 8c, top and Supplementary Movie 11) to form a

bi-wheel motor, which exhibits rolling and steering capability upon controllable IR light illumination. Specifically, symmetric illumination along the middle axis will cause straight motion and illumination on one side will result in left or right turning (Supplementary Fig. 20), respectively. Supplementary Fig. 21 shows the variations of the displacement and rotation angle with the time of a bi-wheel motor in three different moving modes: left turning, right turning, and straight rolling. The bi-wheel motor exhibits excellent straight-rolling motion indicated by little displacement along the lateral direction and small fluctuation of rotation angle. Meanwhile, the bi-wheel exhibits a good turning ability as proven by the ever-increasing displacement along the lateral direction and rotation angle. Based on the developed bi-wheel system, a four-wheel chassis system is developed which can achieve more versatile locomotion under the driving by the IR illumination (Fig. 8c, middle and Supplementary Movie 12). Upon illumination from one side, the four-wheel chassis is able to move straightly away from IR light, which shows robust locomotive ability. A four-wheel rolling truck can be developed by a parallel assembly of four rolls and shows stable locomotion ability under lateral IR light illumination (Fig. 8c, bottom and Supplementary Movie 13).

## Discussion

We present a facile and cost-effective strategy for the development of programmable untethered soft robotics, including thermal-induced morphing systems as soft actuators and light-driven motors, by using SGA/PE bilayer films on the basis of

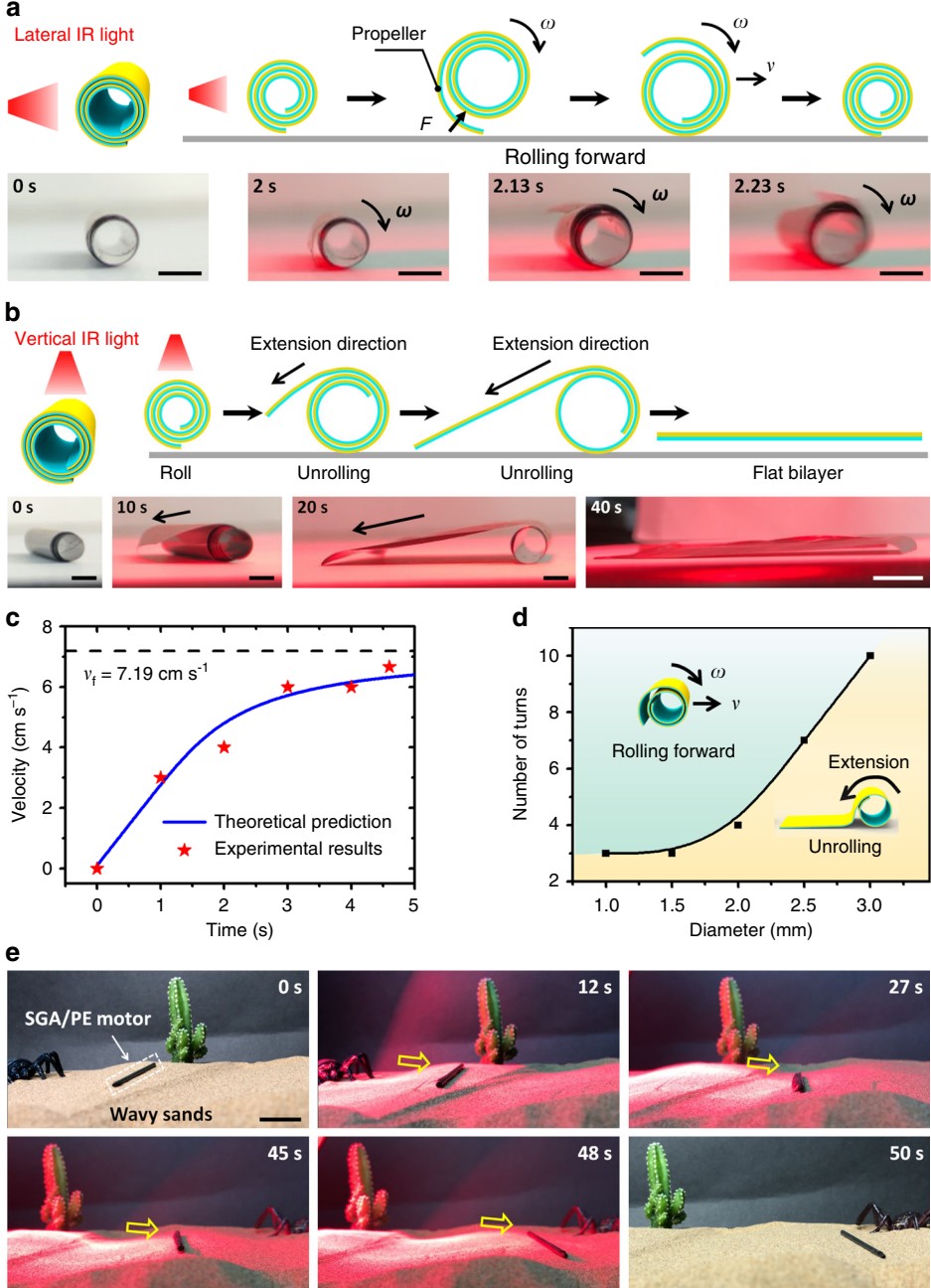

**Fig. 7 Rolling motor based on SGA/PE roll. a** Lateral IR light illumination could cause rolling locomotion of an SGA/PE roll (Supplementary Movie 7). **b** Vertical illumination can make the SGA/PE roll unfold. **c** Evolution of the rolling speed with the time of a rolling motor. **d** An SGA/PE roll may roll or not under lateral IR light illumination depending on its diameters and number of turns. **e** Snapshots of a rolling motor moving on wavy sands (Supplementary Movie 8). Scale bars: 2 mm (**a**); 2 mm (**b**, the left three optical images); 10 mm (**b**, the last optical image); 2 cm (**e**).

our findings of the unique asymmetric elastoplasticity of the SGA under tension and compression. Like many bilayer films, the as-prepared SGA/PE film exhibits reversible bending and coiling behavior in response to the variation of temperature. What is unique is that when an as-prepared SGA/PE bilayer is tempered in a constrained space, the residual stress developed inside will make it to assume another morphology at the ambient temperature without sacrificing its thermal-induced morphing aptitude. This property of the SGA/PE bilayer films is essentially ascribed to the asymmetric elastoplastic behavior of the SGA under tension and compression and thus endows us with the flexibility to program the initial configurations of the thermal-induced morphing systems by applying various

strategies such as patterned SGA patches, nonuniform SGA films, and localized tempering. Thanks to such unique morphing behaviors, the SGA/PE bilayer film is applied to develop a variety of thermal-responsive or light-responsive actuators including sit-up robots, artificial iris, and artificial water lily, etc. More importantly, the tempered SGA/PE bi-material, which spontaneously curls into a roll, can achieve rolling locomotion under lateral infrared lighting, yielding an untethered light-driven motor. The results of our work not only demonstrate an alternative strategy for creating untethered soft robots, artificial muscles, and reconfigurable devices but also provide a philosophy for fabricating 2D material-based smart materials and structures.

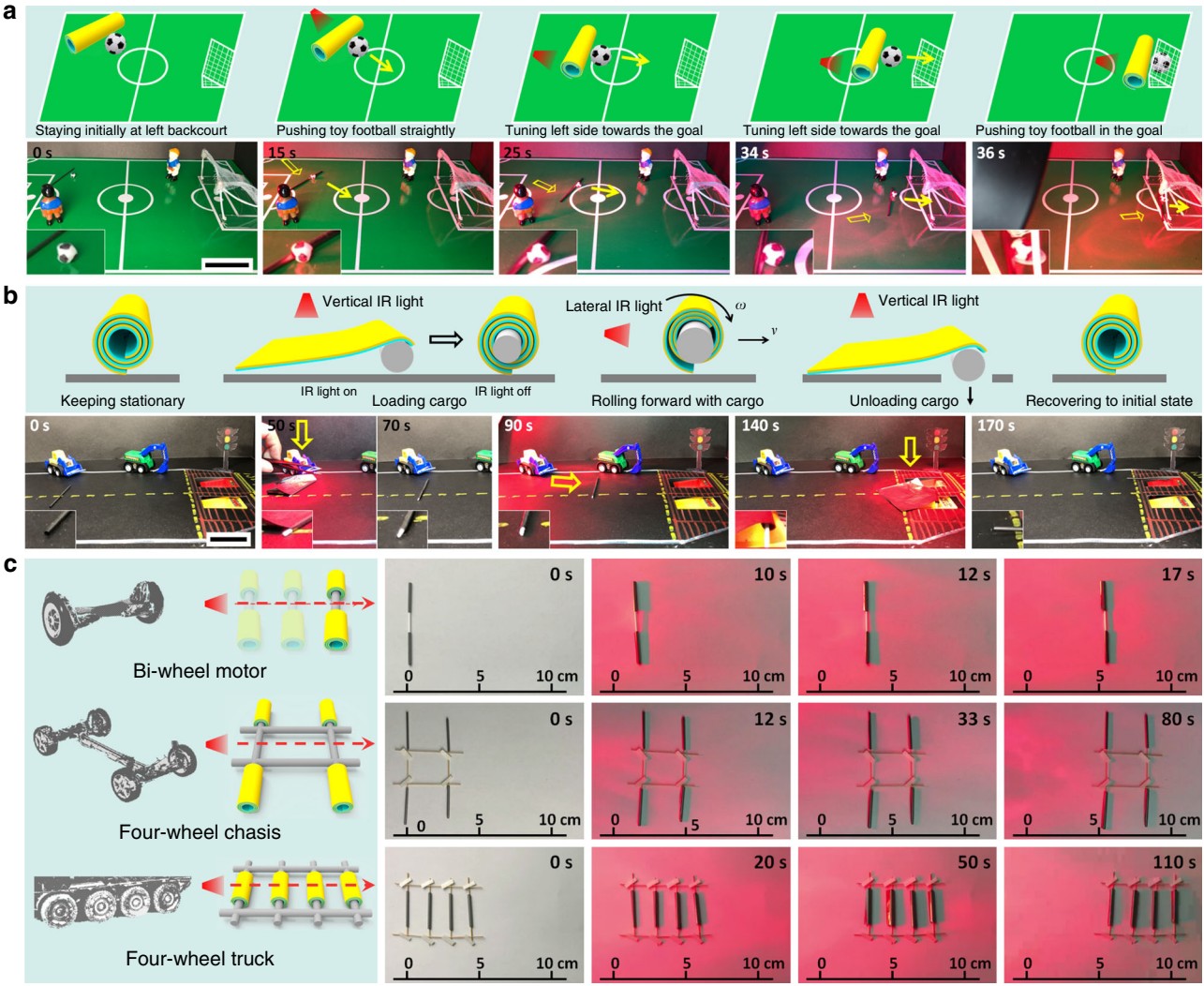

**Fig. 8 Rolling motors for mechanical energy output applications and multiple-wheel assembly systems. a** Controlling locomotion of a light-driven motor (SGA/PE roll) under lateral IR light to push a toy football into the goal of an artificial football field (Supplementary Movie 9). **b** Loading, transporting, unloading of cargo by an SGA/PE roll under controlled IR light illumination (Supplementary Movie 10). **c** A bi-wheel motor assembled from two SGA/PE rolls (Supplementary Movie 11), a four-wheel chasis system by a parallel assembly of two bi-wheel motors (Supplementary Movie 12) and a four-wheel truck by the parallel assembly of four rolls with a home-made frame showing robust locomotion ability upon lateral IR light illumination (Supplementary Movie 13). Scale bars: 5 cm (**a**, **b**).

## Methods

**Fabrication of SGA/PE bilayer film**. The graphene flakes are fabricated using the solvent-assisted mechanical exfoliation method[64]. The preparation of the SGA film is carried out according to the Langmuir–Blodgett method described in our previous study[48–50]. First, few-layer graphene flakes are dispersed into anhydrous ethanol solution (0.5 mg mL$^{-1}$), followed by ultrasonication for 3 h. Then, the ethanol-assisted graphene dispersion is sprayed onto the water surface in a container, resulting in a layer of graphene flakes floating on the water surface. A piece of sponge block is gradually inserted into the water from aside, resulting in the breakage of equilibrium in the surface tension of the air/water interface. The floating graphene flakes then recede away from the sponge block and aggregate to form a condensed SGA layer. Successive pieces of sponge blocks can be used until the graphene flakes cannot be condensed further. A piece of commercial plastic wrap (low-density polyethylene) is fixed on a glass slide with the assistance of ethanol for ensuring smooth and close contact with the slide surface. The SGA film floating on air/water interface is then transferred to the PE film (60 mm × 60 mm) using the lift-up transferring method, followed by a drying procedure with nitrogen gas at room temperature (25 °C). A multi-ply SGA film (1, 2, 3, 4, 5, and 6 plies) can be prepared simply by repeating the transferring and drying process for multiple times. Finally, the dried SGA/PE bilayer is ready for use after being sliced into small strips of specific dimensions. Discrete SGA patches are realized by mask method: discrete areas of the PE film are pre-covered by PE strips with specific dimensions. Then the PE film with masks is used to transfer SGA films. Finally, the SGA film on the mask areas is removed by peeling off the masks, giving rise to the discrete SGA patches on PE film. For gradient SGA patches, SGA films are transferred in a non-uniform way to the PE films. Specifically, SGA films are transferred to only end parts of PE films in the first transferring process. Then SGA films are transferred to a larger area (including the area in the first transferring process) of PE films in the next transferring process. Repeating this process, gradient SGA patches are obtained on the PE films.

**Constrained tempering of SGA/PE bilayer film**. An as-prepared flat SGA/PE strip is sandwiched by two pieces of glass slide to suppress the possible bending deformation and then placed on a hotplate with a preset temperature for heating. After 10 min, the sample is removed from the hotplate and cools down to room temperature in the ambient environment. The constraint on the bilayer strip is released by removing the glass slides, resulting in a tempered SGA/PE bilayer film. For localized tempering, localized heating is realized by area heat source of a hotplate or point heat source of a red laser beam (ZengYuanDa high-power laser flashlight, wavelength: 650 nm). Specifically, only one side of the SGA/PE films is placed on the hotplate for localized tempering on side parts of SGA/PE films. Sliced glass slides with a specific width are stacked on a hotplate to provide local heating for localized tempering in the middle parts of SGA/PE films. Localized tempering by the laser beam is realized by direct writing on the SGA/PE films with constraint.

**Actuation test of SGA/PE bilayer film**. SGA/PE rolls are placed on the hotplate at preset actuation temperature ($\Delta T$) for thermal-induced unrolling deformation. Then the actuated flat SGA/PE film is removed from the hotplate for observation of the shape recovery process under cooling. This process is repeated several times to

confirm the good durability of the reversible actuation performance. The whole process is recorded by a digital camera. Infrared (IR) light lamp (Philips infrared light lamp R95, 100 W) is used for light-driven shape morphing devices and rolling motors.

**Numerical simulations**. MD simulations are performed to determine the mechanical behaviors of the SGA layer under uniaxial tension and compression with the LAMMPS packages[65]. The SGA layer is idealized as a multilayer architecture of periodically stacked graphene flakes. Each flake consists of three graphene layers with a length of 99 nm in the longitudinal direction. A representative volume element (RVE) with two stacked flakes is adopted to represent the whole structure. The overlap length between the adjacent flakes is 98 nm. The size of the RVE cell along the $x$-direction, $y$-direction, and $z$-direction are 100 nm, 6 nm, and 2.1 nm, respectively. Periodic boundary conditions are applied along each direction of the simulation cell. A well-developed coarse-grain (CG) molecular model of graphene is adopted with parameter settings based on the published reference[66]. And the interlayer interaction is described by Lennard–Jones (L–J) potential $V(r) = 4\varepsilon\left[\left(\frac{\sigma}{r}\right)^{12} - \left(\frac{\sigma}{r}\right)^{6}\right]$ with parameters $\sigma = 3.46$ Å and $\varepsilon = 0.21$ kcal mol$^{-1}$.

Finite element analysis (ABAQUS, Dassault Systèmes) is carried out to simulate the morphing behavior of the SGA/PE bilayer film through tempering with constraint and the related actuators in response to temperature variation. The PE layer is modeled as a purely elastic material with Young's modulus 300 MPa[61] and Poisson's ratio 0.46[62]. The SGA layer is assumed as a continuum material with asymmetric elastoplastic behavior under tension and compression, namely elastic and ideally plastic under tension and purely elastic under compression. Based on the MD simulation results (Fig. 4a), the elastic modulus and yielding strength under tension are taken as 20.7 GPa and 20.4 MPa, respectively, and the elastic modulus under compression is taken as 2.2 GPa. The Poisson's ratio of the SGA layer is taken as 0.19, which is calculated from MD simulation. The linear thermal expansion coefficients of the PE and SGA are taken as $\alpha = 4 \times 10^{-4}/{}^\circ\text{C}$[53] and zero, respectively. Perfect bonding is assumed between the SGA and PE layers.

**Structural and material characterizations**. A field-emission scanning electron microscope (Hitachi S4800) is used to investigate the morphology of SGA and PE film. The Raman scattering measurements are performed at room temperature on a Raman system (in Via-reflex, Renishaw) with confocal microscopy. The solid-state diode laser (532 nm) is used as an excitation source with a frequency range of 3200-1000 cm$^{-1}$. The uniaxial tensile test is carried out with a universal material testing machine (Instron 5567). The 2D X-ray diffraction (XRD) pattern of the PE film is recorded using Bruker D8 Discover with Cu-Kα radiation ($\lambda = 1.5418$ Å). The distance between samples to the detector is fixed at 10 cm. The intensity of IR light illumination is characterized by IR power meter (LH-129). IR cameras (FLIR C2 and E8) are employed to realize a real-time recording of the surface temperature and IR images.

## Data availability
The data that support the findings of this study are available from the corresponding authors upon reasonable request. See Author Contributions for the responsible persons for specific data sets.

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

## Acknowledgements

This work was supported by the Natural Science Foundation of China (51803226, 51573203, and 11772283), the Bureau of International Cooperation of Chinese Academy of Sciences (174433KYSB20170061), Key Research Program of Frontier Science, Chinese Academy of Sciences (QYZDB-SSW-SLH036), NSFC-Zhejiang Joint Fund for the Integration of Industrialization and Informatization (U1909215), Postdoctoral Innovation Talent Support Program (BX20180321), China Postdoctoral Science Foundation (2018M630695), General Research Fund (PolyU 152064/15E, PolyU 5293/13E) from Hong Kong RGC, and Ningbo Scientific and Technological Innovation 2025 Major Project (2018B10057).

## Author contributions

S.W., P.X., H.Y., and T.C. conceived the concepts and designed the research; S.W., P.X., Y.L., and W.L. performed the fabrications, characterizations and static morphing demonstrations; S.W., C.C, C.Z., and G.Y. performed motor experiments; H.Y., Y.G., and A.W. conducted the theoretical modeling, carried out the finite element analysis and molecular dynamics simulations; S.W., H.Y., Y.G., A.W., P.X., and T.C. wrote the paper. All authors discussed the results and commented on the manuscript.

## Competing interests

The authors declare no competing interests.
