## [Peer Review File · Nature Communications]

Reviewers' Comments:

Reviewer #1:

Remarks to the Author:

please see attached in step 4

[Comment from the Editor: Please see attached Pdf]

Reviewer #2:

Remarks to the Author:

The authors prepared a bilayer film of SGA/PE which exhibits swift morphing behavior in response to the variation of the surrounding temperature. The manuscript should be revised before acceptance. The detailed comments are as follow:

1. The as-prepared bi-material film shows bending and coiling behavior in response to the external temperature variation. The conditions for bending or coiling need to be clarified. Is the size of film the most important factor affecting actuation performance? Are there other factors?
2. P4 line 115, is the graphene reduced graphene oxide or CVD based graphene? If the graphene used here is home-synthesized, the preparation method needs to be provided.
3. Could you show the DSC curves of PE and SGA/PE film, which are important for understanding the mechanism of the morphing behavior of the film?
4. The sizes of the SGA/PE film in Fig.3 and in Fig. 8 are different. Could the author show the influence of size and the ratio of SGA and PE on the morphing behavior?
5. The film showed inconsistent colors in different figures, e.g. gray appearance in Fig. 2B, Fig. 3A, Fig, 3B, while black color in Fig. 3E, Fig. 5G, Fig. 6). What is cause for this difference?
6. Why does the graphene layer, instead of the PE layer, swell with laser writing in Figure 5G
7. The author said: "as-prepared SGA/PE bilayer is pre-treated with a heating and a subsequent cooling process (similar to the tempering process in metallurgy) in a constrained space." An as-prepared flat SGA/PE strip is sandwiched by two pieces of glass slide to suppress the possible bending deformation.
Glass slide can only constrain the film in the vertical direction. What may happen if the film deformation is constrained in both vertical and parallel directions?
8. Do IR light and laser illumination cause difference temperatures of the film? Does the illumination time affect the temperature of the film?
9. How is the structural stability of the bi-layer film, e.g, does delamination or exfoliation occur during the deformation?

Reviewer #3:

Remarks to the Author:

In this work, the authors report the graphene/polyethylene based bimorph soft actuators and demonstrate their applications in 3D reconfiguration and rolling motors. The working mechanism (i.e., thermal and photothermal actuation) is already well known and extensively demonstrated in other research papers. It is not new to make soft actuators that can bend toward the material with a higher coefficient of thermal expansion by introducing thermal stress (Adv. Mater. 2015, 27, 7867-7873). Also, the correlation between the so-called asymmetric elastoplasticity and the actuator performance is too speculative. Thus, I do not suggest the publication of the manuscript in Nature Communications.

Manuscript: NCOMMS-20-06504

Type: Article

Title: Asymmetric elastoplasticity of stacked graphene assembly actualizes programmable untethered soft robotics

Authors: Shuai Wang, Yang Gao, Anran Wei, Peng Xiao, Yun Liang, Wei Lu, Chinyin Chen, Chi Zhang, Guilin Yang, Haimin Yao, Tao Chen

Summary: The authors use the Langmuir-Blodgett method to deposit graphene layers on polyethylene. These SGA-PE layers exhibit morphing behaviors if exposed to temperature (~50°C). The tempering (heat treatment under pressure) of SGA-PE layers enable new types of morphing behaviors. Previously, SGA-PE layers flattened at room temperature and deformed at increased temperatures; in this work, after tempering, SGA-PE layers deform at room temperatures and flatten at increased temperatures. This work essentially improves on the types of folding mechanisms that are possible with graphene coated polymers.

I recommend that this paper should be considered for publication after major revisions (i.e. restructuring and rewriting of the manuscript, additional demonstrations that indicate the benefits of this material (using tempering) relative to those that exist, and a study on actuation cycles and material fatigue). The authors dilute the novelty of their work by showing demonstrations of mechanisms that have been previously reported (which they missed citing [1]). They fail to effectively extract the novelty of their work, that is, the use of tempering on SGA/PE bilayers for the creation of new types of actuation modes; in the current state, the manuscript is difficult to follow and many of their demonstrations do not support the narrative of their work.

Major Issues:

- Given the title of the manuscript, this work insufficiently addresses the field of soft robotics. If these actuators are meant to be used for robotics applications, we will also need to know more details about their performance such as the number of actuation cycles before they experience material fatigue. Please include an extensive study on the actuators. How much force are the actuators capable of exerting? What is your success rate of building them? Do 10 out of 10 actuators work? The authors also talked about cost-effectiveness without addressing what that actually means (in \$). How expensive is a single actuator (material cost and time to fabricate a single device (trained/untrained students)? What keeps the SGA and PE layers attached to each other (van der Waals forces)? How robust are your actuators (could you step onto them and would they still work)?
- The authors spend significant amount of text on explaining and demonstrating SGA/PE layers without heat tempering (artificial mimosa, sit-up robot), however, this does not seem the novelty of their work. The layering of graphene onto polyethylene has essentially been done before and its response to heat (light) documented (see reference [1]). The authors missed citing key literature. By doing so, the entire manuscript becomes very difficult to comprehend since the novelty of their work is diluted amongst existing approaches. This manuscript must be rewritten solely focusing on heat tempering with only a brief introduction to SGA/PE bilayers (unless the authors believe their SGA/PE

layering is more advantageous (for soft robotics) if compared to existing work, which they have to prove).

Minor Issues:

- Figure 1 is confusing; E) the authors mention that heat deforms the bilayer to its flat state, however, they indicate a barely unrolled device in the figure. Does your tempered device fully unroll? G) The artificial mimosa and sit-up robot don't take advantage of the constrained tempering, hence, dilute the work. These demonstrations should be taken out and replaced by new experiments that take advantage of the novelty of this work, that is, the heat tempering of SGA/PE bilayers.
- Subfigures 3A and 3B are inconsistent; the right part of Figure 3B should also indicate experimental data instead of schematics.
- Figure 5 shows the temperature-induced self-folding. This work has essentially been done before. If you want to include it, tell us how it is different to existing work [1]. Maybe you could show how the entire cube collapses when applying heat? Can you think of applications where this is actually useful? G) Why is the lasering done sequentially (one edge at the time)? It seems like the laser should program the material all at once when it is sandwiched (constrained) between the tempering layers.

General Comments:

- The novelty of your work has to be clearly stated in light of existing literature. What makes your work novel? In which matrix (e.g. actuation speed?) does your work outperform existing approaches? Why should anyone care about this (in the soft robotics community)? Maybe because the results are easy to reproduce, and the actuators are low-cost and reliable?

References:

1. Xu, Weinan et al., Ultrathin thermoresponsive self-folding 3D graphene, Science advances, 3(10), 2017

Response to Reviewers' Comments for manuscript

NCOMMS-20-06504: "Asymmetric elastoplasticity of stacked graphene assembly actualizes programmable untethered soft robotics"

Reviewer #1

Summary: The authors use the Langmuir-Blodgett method to deposit graphene layers on polyethylene. These SGA-PE layers exhibit morphing behaviors if exposed to temperature (~50°C). The tempering (heat treatment under pressure) of SGA-PE layers enable new types of morphing behaviors. Previously, SGA-PE layers flattened at room temperature and deformed at increased temperatures; in this work, after tempering, SGA-PE layers deform at room temperatures and flatten at increased temperatures. This work essentially improves on the types of folding mechanisms that are possible with graphene coated polymers. I recommend that this paper should be considered for publication after major revisions (i.e. restructuring and rewriting of the manuscript, additional demonstrations that indicate the benefits of this material (using tempering) relative to those that exist, and a study on actuation cycles and material fatigue).

Response: We appreciate your positive comments. The point-to-point responses according to your kind suggestions have been addressed substantially in the revised version of our manuscript.

The authors dilute the novelty of their work by showing demonstrations of mechanisms that have been previously reported (which they missed citing [1]). They fail to effectively extract the novelty of their work, that is, the use of tempering on SGA/PE bilayers for the creation of new types of actuation modes; in the current state, the manuscript is difficult to follow and many of their demonstrations do not support the narrative of their work.

Response: Thanks for your valuable comments. To emphasize the novelty of this work which is the use of tempering on SGA/PE bilayers for the creation of new types of actuation modes. We made a significant revision to the manuscript by reducing the discussion on the SGA/PE layers without heat tempering.

Major Issues:

Given the title of the manuscript, this work insufficiently addresses the field of soft robotics. If these actuators are meant to be used for robotics applications, we will also need to know more details

about their performance such as the number of actuation cycles before they experience material fatigue.

Response: Thanks for raising this issue. To test the stability of the actuation performance of the SGA/PE bilayer film, a long-term actuation test was carried out using cyclic IR irradiation with intensity of 120 mW cm^{-2} . In each cycle, the film was exposed to 5 s of IR illumination, followed by 5 s pause of illumination. Each cycle lasted for 10 s and the entire test of 1000 cycles took $\sim 2.8 \text{ h}$. The curvatures of the film before and after actuation were measured every 20 cycles, as shown in Figure R1a below. It can be seen that the unrolling-rolling actuation of the film exhibits high reversibility and stability even after 1000 cycles. No noticeable deterioration is observed in the actuation performance of the film (Figure R1b, Supplementary Movie 1). This information has been added into the revised manuscript (Line 198-200 on page 8) and revised Supplementary Information (Supplementary Figure 8 on page 14, Supplementary Movie 1).

Figure R1. Stability test of the SGA/PE film upon cyclic IR light illumination. (a) The curvatures of the film before and after 20 cycles of unrolling-rolling actuation. (b) Optical images showing the morphologies of the film after the cycle 1 and cycle 1000, respectively. The irradiation intensity of the IR light is 120 mW cm^{-2} . Scale bars: 5 mm.

Please include an extensive study on the actuators. How much force are the actuators capable of exerting?

Response: Thanks for this constructive comment. To evaluate the force exerted by the actuator, a weightlifting experiment was conducted, as shown in Figure R2. Our results show that the actuator can lift up a load of ten times of the actuator weight upon controlled IR light illumination with intensity of 120 mW cm^{-2} . This information has been added into the revised manuscript (Line 203-205 on page 8) and revised Supplementary Information (Supplementary Figure 10 on page 15).

Figure R2. Schematic illustration and snapshots of the weightlifting experiment under controlled IR light illumination with intensity of 120 mW cm^{-2} . The load is ten times of the actuator weight. Vertical displacement is 8 mm. The actuator here is SGA (6 plies)/PE tempered at $\Delta T = 50 \text{ }^\circ\text{C}$.

What is your success rate of building them? Do 10 out of 10 actuators work?

Response: The success rate of building the SGA/PE actuators in our lab is almost 100%. 10 out of 10 actuators can work with proper working conditions.

The authors also talked about cost-effectiveness without addressing what that actually means (in \$). How expensive is a single actuator (material cost and time to fabricate a single device (trained/untrained students)?

Response: Thanks for raising this practical comment. Cost-effectiveness is one of the advantages our SGA/PE system possesses. The fabrication process does not involve specific and expensive facilities. The consumables and materials applied include pure ethanol, few-layer graphene flakes and PE film. The material cost for a piece of SGA/PE bilayer film is around $\$0.0024/\text{cm}^2$ ($\$0.0018$ for few-layer graphene flakes, cost of PE film can be neglected, $\$0.0006$ for pure ethanol). The time to fabricate a piece of SGA/PE film is about 0.5 h for a trained student, which includes the time for preparing graphene film with a ready graphene dispersion and transferring the SGA film to PE substrate. Therefore, the convenient fabrication process and very low materials cost indicate the cost-effective features of the fabrication process in this work. This point has been added to the revised manuscript (line 123 on page 5).

What keeps the SGA and PE layers attached to each other (van der Waals forces)?

Response: The fabrication process of SGA/PE bilayer films does not involve application of other chemicals. The bilayer film is prepared by transferring SGA film floating on air/water(pure) interface to the PE film using the lift-up transferring method, followed by a drying procedure with nitrogen gas at room temperature. Therefore, it is the van der Waals force that keeps the SGA and PE layers attached to each other.

How robust are your actuators (could you step onto them and would they still work)?

Response: Thank you for the constructive comment. To testify the robustness of the SGA/PE actuators, trampling tests were carried out over a tempered SGA/PE film with pre-designed coiling configuration, as shown in Figure R3. After being trampled along the film thickness direction, the film shows little change in its actuation performance in comparison to the intact one (Figure R3b, c). Another trampling is conducted along the side direction (Figure R3d). It can be seen that the morphing behavior of the actuator is still preserved even though its pre-designed shape cannot be maintained because trampling causes permanent deformation to the actuator. This information has been added into the revised manuscript (Line 200-201 on page 8) and revised Supplementary Information (Supplementary Figure 9 on page 15).

Figure R3. Robustness of the SGA/PE film actuators. (a) Optical images showing the actuator being trampled by an adult (~53 kg). (b) Actuation performance of the intact SGA/PE film (c), actuation performance after the first trampling along the film thickness direction, and (d) actuation

performance after the second trampling along the side direction of a tempered SGA/PE actuator. Scale bar: 5 mm.

The authors spend significant amount of text on explaining and demonstrating SGA/PE layers without heat tempering (artificial mimosa, sit-up robot), however, this does not seem the novelty of their work. The layering of graphene onto polyethylene has essentially been done before and its response to heat (light) documented (see reference [1]). The authors missed citing key literature. By doing so, the entire manuscript becomes very difficult to comprehend since the novelty of their work is diluted amongst existing approaches. This manuscript must be rewritten solely focusing on heat tempering with only a brief introduction to SGA/PE bilayers (unless the authors believe their SGA/PE layering is more advantageous (for soft robotics) if compared to existing work, which they have to prove).

Response: Thanks for raising this constructive comment. We agree with this comment and therefore greatly diminished the discussion on the SGA/PE without tempering in the revised manuscript especially in Fig. 1 and Fig. 6. But we keep the sit-up robot because its fabrication does involve constrained tempering with direct laser writing. Moreover, we added the key reference (*Sci. Adv.* **3**, e1701084 (2017)) in the revised manuscript (Refs. 12 in the reference list). We believe these revisions will make the novelty of this work clearer and focused. The revisions made in the revised manuscript including text modifications (Line 88-89 and 94-96 on page 3; Line 293-294, 303-305 on page 12; Line 387-391 and 397 on page 17), Fig.1 (page 4) and Fig. 6 (page 13). A new experiment using heat tempering has also been added into the revised manuscript (Line 303-305 on page 12) and revised Supplementary Information (Supplementary Fig. 17 on page 19).

Fig. 1 Illustration of programmable thermal-induced morphing systems based on SGA/PE bilayer films. **a-c** Schematics showing the process of constrained tempering on SGA/PE films. **d, e** Schematics showing the thermal-induced shape morphing of a pristine SGA/PE film and a tempered SGA/PE film respectively. **f** Schematic illustration of the deformation mechanism accounting for the asymmetric elastoplasticity of the SGA under tension and compression. **g** Three representative shape morphing systems with programmable initial configurations made of SGA/PE bilayers. **h** SGA/PE bilayer roll as a light-driven rolling motor.

Fig. 6 Typical morphing systems assembled from basic morphing units of SG/PE bilayer. **a** Schematic illustration of the assembly units of an artificial water lily through constrained tempering. **b** An artificial water lily assembled from the curved units blooms under sunlight (20 mW cm^{-2}) (Supplementary Movie 4). **c** Schematic illustration of the assembly units of sit-up robot through constrained tempering. **d** A simple robot assembled from folded units sits up under IR light illumination (Supplementary Movie 6).

Minor Issues:

Figure 1 is confusing; E) the authors mention that heat deforms the bilayer to its flat state, however, they indicate a barely unrolled device in the figure. Does your tempered device fully unroll?

Response: Thanks for your suggestive comment. The tempered device can fully unroll under certain actuating temperature. Figure R4 shows the variation of the curvature of a tempered device with increased actuating temperature. The insets show that the tempered device (initial curvature is 1.85 mm^{-1}) eventually unrolls to the flat state at temperature increment of $40 \text{ }^\circ\text{C}$. Figure 1 has been

revised to address this issue. This information has been added into the revised manuscript (Line 173 on page 7, line 192 on page 8) and revised Supplementary Information (Supplementary Figure 5 on page 13).

Figure R4. Variation of the curvature of a tempered SGA/PE film with actuating temperature. Scale bar: 3 mm.

G) The artificial mimosa and sit-up robot don't take advantage of the constrained tempering, hence, dilute the work. These demonstrations should be taken out and replaced by new experiments that take advantage of the novelty of this work, that is, the heat tempering of SGA/PE bilayers.

Response: Thanks for this constructive comment. We have taken out the artificial mimosa, as shown in the revised Fig. 1 and Fig. 6. But we keep the sit-up robot demonstration as its fabrication involves constrained tempering with laser writing. A new experiment that takes advantage of the heat tempering has been added, as shown in Figure R5. This information has been added into the revised manuscript (Line 303-305 on page 12) and revised Supplementary Information (Supplementary Fig. 17 on page 19).

Figure R5. Tempered SGA/PE bilayers for information storage, encryption, and decryption. (a) Schematic illustration showing a message concealed in a coiled SGA/PE bilayer roll. The thin roll can then be put into a container with a narrow neck for storage. The message can be reversibly displayed and hidden upon the application of external IR light illumination. (b) Optical image of an as-prepared flat SGA/PE film with message written on SGA side. (c) The coiled SGA/PE roll after constrained tempering can pass through a narrow neck of a container. Reversible encryption (d) and decryption (e) of the message under controlled IR light illumination. Scale bars, 1 cm.

Subfigures 3A and 3B are inconsistent; the right part of Figure 3B should also indicate experimental data instead of schematics.

Response: Thanks for this constructive suggestion. Subfigures 3A and 3B have been revised to address this issue as shown in Fig. R6.

Figure R6. (a) Schematic illustration and optical images showing reversible morphing behavior of a pristine flat SGA/PE film upon heating and cooling. (b) Schematic illustration and optical images of constrained tempering on an SGA/PE bilayer and the reversible morphing behavior exhibited by the tempered SGA/PE bilayer in response to thermal stimuli. Scale bars: 5 mm.

Figure 5 shows the temperature-induced self-folding. This work has essentially been done before. If you want to include it, tell us how it is different to existing work [1]. Maybe you could show how the entire cube collapses when applying heat? Can you think of applications where this is actually

useful? G) Why is the lasering done sequentially (one edge at the time)? It seems like the laser should program the material all at once when it is sandwiched (constrained) between the tempering layers.

Response: In the existing work mentioned [1] (*Sci. Adv.* **3**, e1701084 (2017)), the 3D structure (cube) was obtained and maintained under external thermal stimuli. In contrast, the cube shown in this work was obtained by constrained tempering with direct laser writing and therefore it can be maintained at ambient temperature. This is the significant difference between them. The collapse of our cube upon heating is shown in Figure R7. The potential applications of such programmable responsive deformation include drug releasing, biomimetic soft robotics, and artificial muscles. Sequential tempering is not necessary for manufacturing our cube. One can apply tempering all at once to fabricate the cube. In our work, sequential tempering was applied for the purpose of showing the competence of constrained tempering of SGA/PE in shape programming. This point has been added into the revised manuscript (Line 287-289 on page 12) and revised Supplementary Information (Supplementary Figure 14 on page 17).

Figure R7. The cube keeps its configuration at room temperature (a) and unfolds upon heating ($\Delta T = 10\text{ }^{\circ}\text{C}$) (b). Scale bar: 10 mm.

General Comments: The novelty of your work has to be clearly stated in light of existing literature. What makes your work novel? In which matrix (e.g. actuation speed?) does your work outperform existing approaches? Why should anyone care about this (in the soft robotics community)? Maybe because the results are easy to reproduce, and the actuators are lowcost and reliable?

Response: Thanks for the valuable comment. The novelties and advantages of our actuators, in comparison to the existing counterparts, can be summarized as follows:

1. High programmability of the initial configuration/shape of the actuators: For the traditional actuators reported in previous works (*Adv. Mater.* **27**, 7867-7873 (2015); *Chem. Mater.* **29**,

9793-9801 (2017); *Nat. Commun.* **9**, 4148 (2018); *Sci. Adv.* **5**, eaaw7956 (2019); *Adv. Mater.* **31**, e1808235 (2019)), the initial shape is normally fixed and cannot be programmed. The deformation is triggered and maintained upon the provision of external stimuli; while the shape of our SGA/PE-based actuator can be programmed as needed and maintained at ambient temperature.

2. High actuation performance: The morphing speed (0.089 s^{-1}) of the SGA/PE-based actuators is comparable to the state-of-the-art actuators (Fig. 3g).
3. Locomotion rather than actuation: Most previous bilayer-based actuators can only achieve deformation and actuation on a fixed position. Our SGA/PE can be used to achieve long-distance locomotion in addition to actuation.
4. Low cost and facile fabrication procedure: The SGA/PE bilayer film costs only $\sim \$0.0024/\text{cm}^2$ and the fabrication method is easy to follow with no special and expensive facilities required.
5. Untethered actuation and control: With the SGA/PE bilayer film, light-induced actuation can be achieved and controlled without using cables or wires for provision of power and controlling signals.

Therefore, our work not only present an alternative bilayer-based actuator but also bring novel and additional features with less cost. The points and revisions are presented in the revised manuscript (Line 44-49 and 61-63 on page 2; Line 71-75, 88-90 and 99-101 on page 3; Line 123 on page 5).

Reviewer #2

The authors prepared a bilayer film of SGA/PE which exhibits swift morphing behavior in response to the variation of the surrounding temperature. The manuscript should be revised before acceptance.

Response: Thank you for your positive comments. We have modified the manuscript substantially according to your comments. The point-to-point responses are addressed as follows.

The detailed comments are as follow:

1. The as-prepared bi-material film shows bending and coiling behavior in response to the external temperature variation. The conditions for bending or coiling need to be clarified. Is the size of film the most important factor affecting actuation performance? Are there other factors?

Response: Thanks for raising this constructive comment. The conditions for bending or coiling have been clarified in the revised manuscript. To investigate the effect of the dimensions of the films on actuation performance, we have conducted additional experiments with samples of different dimensions. First, we fixed the length of the bilayer films as 5 mm and changed the width from 1 mm to 5mm. Figure R8a shows that coiling curvature of these samples in response to a temperature increase ($\Delta T=15$ °C) exhibits slight variation from 0.67 mm^{-1} (width, 1 mm) to 0.60 mm^{-1} (width, 5 mm). Then, we kept the aspect ratio of the films fixed and equal to 2.0 and changed the width from 1 mm to 5mm. The coiling curvature of these batch of samples in response to a temperature increase ($\Delta T=15$ °C) stays at 0.60 mm^{-1} with a little variation, as shown in Figure R8b. Therefore, the size of the films does not affect the actuation performance too much. In addition, we investigated the effects of other two factors, actuating temperatures ($\Delta T=5, 10, 15$ °C) and SGA ply number (1, 2, 3, 4, 5, 6 plies), on the actuation performance of as-prepared bilayer films. The result is shown in Figure R9. Clearly, the curvatures increase significantly with the increase of actuating temperature. As for the effect of SGA ply number, thicker SGA layer promotes the deformation of the bilayer. However, such promotion effect diminishes with the increase of ply number. The above results and related discussions have also been added into the revised manuscript (Line 161-162 on page 6) and revised Supplementary Information (Supplementary Figure 2 on page 11, Supplementary Figure 3 on page 12).

Figure R8. Actuation performance of the as-prepared SGA/PE bilayers with different sizes. (a) The coiling curvature of the samples with different widths and the same length of 5 mm. (b) The coiling curvature of the samples with different widths and the same aspect ratio of 2. The SGA ply number in all samples is 3 and the actuating temperature is $\Delta T=15^\circ\text{C}$. Scale bar: 2 mm.

Figure R9. Actuation performance of as-prepared SGA/PE bilayers with varied SGA ply number upon different actuation temperatures.

2. P4 line 115, is the graphene reduced graphene oxide or CVD based graphene? If the graphene used here is home-synthesized, the preparation method needs to be provided.

Response: The graphene we used was fabricated by solvent-assisted mechanical exfoliation method (*Nat. Nanotechnol.* **3**, 538-542 (2008)). The lateral size of the few-layer graphene flakes is about 5~15 μm and the thickness is 2~7 nm. This information has been added into the revised manuscript (Line 405 on page 17).

3. Could you show the DSC curves of PE and SGA/PE film, which are important for understanding the mechanism of the morphing behavior of the film?

Response: Thermal analysis of PE and SGA/PE film was conducted with a DSC 214 (NETZSCH). Samples were heated (25–200 °C at 10 °C min⁻¹) under nitrogen atmosphere (with a flow rate of 40 ml min⁻¹). The DSC curves of PE and SGA/PE film are shown in Figure R10. The DSC thermogram recorded for PE film shows a double endothermic peak around 111 °C and 123 °C, which is ascribed to the crystalline melting of linear low-density polyethylene (LLDPE)/low density polyethylene (LDPE) blend film. The calorimetric curve shape of SGA/PE film shows great similarity with that of the PE film. The mechanism of the morphing behavior of the as-prepared flat SGA/PE bilayer is attributed to the misfit of thermal expansion between the SGA and PE layers. The thermal expansion coefficient of graphene (*Phys. Rev. Lett.* **106**, 135501 (2011)) is almost neglectable in comparison to that of PE ($4.0 \times 10^{-4}/^{\circ}\text{C}$) (*IEEE Electr. Insul. M.* **27**, 8-16 (2011)). The bilayer film curls in response to temperature change due to the resulting eigenstress on the SGA/PE interface. Under this circumstance, the SGA layer is wrapped inside the PE layer. When the SGA/PE film is pre-tempered in a constrained space (like the tempering process in metallurgy), a totally different morphing behavior is observed. After releasing the constraint, the SGA/PE film curls spontaneously into a roll with PE layer wrapped inside by the SGA layer, which is opposite to that of the as-prepared sample without tempering. Such opposite curling direction can be attributed to the asymmetric elastoplastic property of the SGA layer under tension and compression.

Figure R10. DSC curves of the PE film and SGA/PE film measured at a scanning rate of 10 °C/min.

4. The sizes of the SGA/PE film in Fig.3 and in Fig. 8 are different. Could the author show the influence of size and the ratio of SGA and PE on the morphing behavior?

Response: For the as-prepared SGA/PE films, the size and aspect ratio of the films have little influence on the actuation performance, are shown in Figure R8. For the tempered SGA/PE films, the influence of the size and aspect ratio on the morphing behavior were investigated and the results are shown in Figure R11. In Fig. R11a, the length of the bilayer films is fixed at 5 mm and the width changes from 1 mm to 5mm. In Fig. R11b, the ratio of length to width is fixed at 2 and the width changes from 1 mm to 5mm. The average curvature stays at 1.0 mm^{-1} with $<10\%$ fluctuation, implying that size and aspect ratio of the tempered SGA/PE films have little effect on their morphing behavior. This information has been added into the revised manuscript (Line 172-173 on page 6) and revised Supplementary Information (Supplementary Figure 4 on page 12).

Figure R11. The morphing behavior of tempered SGA/PE films with different sizes. (a) The curvature of tempered samples with different widths and the same length of 5 mm. (b) The curvature of tempered samples with different widths and the same aspect ratio of 2. The SGA ply number in all samples is 3 and the tempering temperature is $\Delta T=30 \text{ }^\circ\text{C}$. Scale bar: 2 mm.

5. The film showed inconsistent colors in different figures, e.g. gray appearance in Fig. 2B, Fig. 3A, Fig. 3B, while black color in Fig. 3E, Fig. 5G, Fig. 6). What is cause for this difference?

Response: Thanks for raising this issue. The inconsistent colors of SGA/PE films are attributed to different appearance of both sides of the film. Figure R12 shows the optical images observed from two different sides of the bilayer film. While the SGA side exhibits gray color, the PE side exhibits black color. This clarification has been added in the revised manuscript (Line 124 on page 5).

Figure R12. Optical images of the SGA/PE bilayer film taken from different sides. Scale bar: 5 mm.

6. Why does the graphene layer, instead of the PE layer, swell with laser writing in Figure 5G

Response: In Figure 5G, it looks like the graphene layer, instead of the PE layer, swells slightly because after a constrained tempering progress with laser, the SGA layer has developed unrecoverable deformation while the PE layer does not. Such behavior can be attributed to the unique mechanical property of the SGA: it exhibits plasticity under tension while high elasticity under compression. The asymmetric elastoplasticity of SGA has been verified in Fig. 4 shown as follows.

Fig. 4 Numerical verification of the asymmetric elastoplasticity of SGA. **a** The loading and unloading stress-strain curves of an SGA layer under uniaxial tension (red curve) and compression (blue curve) as calculated by MD simulation. The black dash lines represent the equivalent stress-strain relations of SGA when treated as a continuum. **b** Schematics of the idealized structure of SGA and the calculated snapshots of the deforming process under tension and compression by MD simulation (see Supplementary Movie 1 for details). **c** Variation of curling curvature (k) of tempered SGA/PE bilayer as a function of SGA layer thickness and tempering temperature as predicted by theory (Eq. (2)) and FEA simulation.

7. The author said: “as-prepared SGA/PE bilayer is pre-treated with a heating and a subsequent cooling process (similar to the tempering process in metallurgy) in a constrained space.” An as-prepared flat SGA/PE strip is sandwiched by two pieces of glass slide to suppress the possible bending deformation. Glass slide can only constrain the film in the vertical direction. What may happen if the film deformation is constrained in both vertical and parallel directions?

Response: Thanks for raising this interesting and instructive suggestion. In the tempering procedure, the as-prepared flat SGA/PE film is sandwiched by two pieces of glass slides in order to constrain the possible bending deformation of the film along the vertical direction. To further constrain the film deformation in the parallel direction, we added a 500 g weight (equivalent to 8 kPa) on the top glass slide to increase the frictional forces to prevent the possible in-plane sliding of the film along the glass surface. After a tempering process (heating and cooling) with $\Delta T = 30\text{ }^{\circ}\text{C}$, the SGA/PE film was released and exhibited curvature only 0.09 mm^{-1} , which is one order of magnitude lower than that in the case without pressure (i.e., 0.99 mm^{-1}), as shown in Figure R13. That is, the tempering effect could be greatly weakened if the film is constrained in the parallel direction. This result also confirms the importance of the asymmetric elastoplasticity of the SGA to the shape programmability of the SGA/PE bilayer. This information has been added into the revised manuscript (Line 193-195 on page 8) and revised Supplementary Information (Supplementary Figure 6 on page 13).

Figure R13. Comparison of the configurations of SGA/PE films after tempering with and without extra vertical pressure. The SGA ply number is 3 and the tempering temperature is $\Delta T = 30\text{ }^{\circ}\text{C}$. Scale bar: 5 mm.

8. Do IR light and laser illumination cause difference temperatures of the film? Does the illumination time affect the temperature of the film?

Response: Yes, IR light and laser illuminations cause different temperature increase on the SGA/PE film. Figure R14a shows the temperature of a SGA/PE film exposed to laser illumination and IR lamp illumination, respectively. It can be seen that the highest temperature caused by laser illumination is around 43 °C, which is higher than that caused by the IR light lamp illumination (32 °C) at the same illumination distance (35 cm). The effect of the illumination time on the temperature of the film is shown in Figure R14b. The temperature increases rapidly in 2 s and then maintains at a stable value (with little fluctuation) upon the subsequent illumination.

Figure R14. Temperature change of the SGA/PE bilayers under IR light lamp and laser illuminations, respectively. (a) Cyclic temperature change versus time curve under IR light lamp and laser illumination at the same illumination distance of 35 cm. (b) Temperature change versus time curve under IR light lamp and laser illumination at illumination distance of 35 cm. The SGA ply number is 6 plies.

9. How is the structural stability of the bi-layer film, e.g. does delamination or exfoliation occur during the deformation?

Response: Thanks for raising this issue. To test the stability of the actuation performance of the SGA/PE bilayer film, a long-term actuation test was carried out using cyclic IR irradiation with intensity of 120 mW cm^{-2} . In each cycle, the film was exposed to 5 s of IR illumination, followed by 5 s pause of illumination. Each cycle lasted for 10 s and the entire test of 1000 cycles took ~ 2.8 h. The curvatures of the film before and after actuation were measured every 20 cycles, as shown in Figure R1a below. It can be seen that the unrolling-rolling actuation of the film exhibits high reversibility and stability even after 1000 cycles. No noticeable deterioration is observed in the actuation performance of the film (Figure R1b, Supplementary Movie 1). This information has been

added into the revised main text (Line 198-200 on page 8) and revised Supplementary Information (Supplementary Figure 8 on page 14, Supplementary Movie 1).

Figure R1. Stability test of the SGA/PE film upon cyclic IR light illumination. (a) The curvatures of the film before and after 20 cycles of unrolling-rolling actuation. (b) Optical images showing the morphologies of the film after the cycle 1 and cycle 1000, respectively. The irradiation intensity of the IR light is 120 mW cm⁻². Scale bars: 5 mm.

Reviewer #3

In this work, the authors report the graphene/polyethylene based bimorph soft actuators and demonstrate their applications in 3D reconfiguration and rolling motors. The working mechanism (i.e., thermal and photothermal actuation) is already well known and extensively demonstrated in other research papers. It is not new to make soft actuators that can bend toward the material with a higher coefficient of thermal expansion by introducing thermal stress (Adv. Mater. 2015, 27, 7867-7873).

Response: Thanks for your comments. We have made substantial revisions to the manuscript based on all the reviewer's comments. The novelties and advantages of our actuators, in comparison to the existing counterparts, can be summarized as follows:

1. High programmability of the initial configuration/shape of the actuators: For the traditional actuators reported in previous works (*Adv. Mater.* **27**, 7867-7873 (2015); *Chem. Mater.* **29**, 9793-9801 (2017); *Nat. Commun.* **9**, 4148 (2018); *Sci. Adv.* **5**, eaaw7956 (2019); *Adv. Mater.* **31**, e1808235 (2019)), the initial shape is normally fixed and cannot be programmed. The deformation is triggered and maintained upon the provision of external stimuli; while the shape of our SGA/PE-based actuator can be programmed as needed and maintained at ambient temperature.
2. High actuation performance: The morphing speed (0.089 s^{-1}) of the SGA/PE-based actuators is comparable to the state-of-the-art actuators (Fig. 3g).
3. Locomotion rather than actuation: Most previous bilayer-based actuators can only achieve deformation and actuation on a fixed position. Our SGA/PE can be used to achieve long-distance locomotion in addition to actuation.
4. Low cost and facile fabrication procedure: The SGA/PE bilayer film costs only $\sim \$0.0024/\text{cm}^2$ and the fabrication method is easy to follow with no special and expensive facilities required.
5. Untethered actuation and control: With the SGA/PE bilayer film, light-induced actuation can be achieved and controlled without using cables or wires for provision of power and controlling signals.

Therefore, our work not only present an alternative bilayer-based actuator but also bring novel and additional features with less cost. The points and revisions are presented in the revised manuscript (Line 44-49 and 61-63 on page 2; Line 71-75, 88-90 and 99-101 on page 3; Line 123 on

Also, the correlation between the so-called asymmetric elastoplasticity and the actuator performance is too speculative. Thus, I do not suggest the publication of the manuscript in *Nature Communications*.

Response: The asymmetric elastoplasticity of SGA has been testified by MD simulations, in which the SGA exhibits high plasticity under tension while high elasticity under compression (Fig. 4a). The plasticity under tension is essentially due to the irreversible sliding between the graphene flakes, while the elasticity under compression results from the reversible rippling-like deformation at the nanoscale (Fig. 4b). On the other hand, we also demonstrated with finite element analysis that the SGA/PE bilayer film could not exhibit coiled morphology after a constrained tempering process if the SGA does not possess asymmetric elastoplasticity. Moreover, we carried out an additional experiment in which SGA/PE bilayer film was tempered with the in-plane deformation is constrained as well. The result (Figure R13) shows that the tempering effect was greatly weakened when the in-plane deformation of the film is prevented, implying that asymmetric elastoplasticity of the SGA is essential for maintain the shape programmability of the SGA/PE bilayer actuator. Results and discussions about the additional experiment have been added into the revised manuscript (Line 193-195 on page 8) and revised Supplementary Information (Supplementary Figure 6 on page 13).

Figure R13. Comparison of the configurations of SGA/PE films after tempering with and without extra vertical pressure. The SGA ply number is 3 and the tempering temperature is $\Delta T=30$ °C. Scale bar: 5 mm.

Reviewers' Comments:

Reviewer #2:

Remarks to the Author:

The manuscript is well revised and can be accepted at current version.

Response to Referees' Comments for manuscript

NCOMMS-20-06504A: "Asymmetric elastoplasticity of stacked graphene assembly actualizes programmable untethered soft robotics"

Reviewer #2

The manuscript is well revised and can be accepted at current version.

Response: We thank the reviewer for accepting this manuscript for publication.